# Autonomous Detection System for Non-Hard-Hat Use at Construction Sites Using Sensor Technology

**Jung Hoon Kim [1], Byung Wan Jo [1,*] , Jun Ho Jo [1], Yun Sung Lee [1] and Do Keun Kim [2]**

[1] Department of Civil and Environmental Engineering, Hanyang University, Seoul 04763, Korea;
kimj32@hanyang.ac.kr (J.H.K.); kpxjuno08@hanyang.ac.kr (J.H.J.); nikeshoo1982@gmail.com (Y.S.L.)

[2] Research and Development Centre, Youngshine D&C, Gyeonggi-do 13487, Korea; kimdokeun@daum.net

\* Correspondence: joycon@hanmail.net; Tel.: +82-2-2220-0327

**Abstract:** In this study, we present a novel method of detecting hard hat use on construction sites using a modified version of an off-the-shelf wearable device. The data-transmitting node of the device contained two sensors, a photoplethysmogram (PPG) and accelerometers (Acc), along with two modules, a global positioning system (GPS) and a low-power wide-area (LoRa) network module. All the components were embedded into a microcontroller unit (MCU) in addition to the power supply. The receiving node included a server that displayed the results via both the Internet of Things (IoT) and smartphones. The LoRa network connected two nodes so that it could function in larger areas such as construction sites at a relatively low cost. The proposed method analyzes the data from a PPG sensor located on the hard hat chin strap and automatically notifies a manager when a worker is not wearing the required hard hat at the site. In addition, by utilizing the PPG sensor data, a heart rate abnormality-detecting feature was added based on an age-adjusted maximum heart rate formula. In validation tests, various PPG sensor locations and shapes were studied, and the results demonstrated the smallest error in the circular shaped sensor located at the upper neck (0.56%). Finally, an IoT monitoring page was created to monitor heart rate abnormalities while identifying hard hat use violations via both PCs and smart phones.

**Keywords:** hard hat; sensor technology; construction; safety; detection; web monitoring

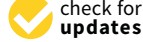



## 1. Introduction

As the quality of life improves, the expectations for and interest in occupational safety are increasing, yet the accident rate in the construction industry has remained the highest among all industries over the past 10 years [1]. Moreover, workers in the construction industry have incurred the largest number of occupational accidents, including injuries and illness [2–7]. Despite global attempts to prohibit construction workers from misusing personal protective equipment (PPE) such as hard hats on the jobsite, this still causes a large proportion of industrial accidents [8–11]. According to the 2019 annual report from the Korea Ministry of Employment and Labor (MOEL), among all industries, 50.1% of fatal accidents such as falls, being struck by an object, and collisions occurred in the construction industry [1]; of these fatal accidents, head injury was found to be the most common cause (41.2%) [12]. Other studies have indicated a high possibility that the head will be the first body part to be impacted from an incident including a fall or being struck by an object on construction sites, and accidents involving head injuries often lead to fatality [12–14]. Moreover, researchers have compared the number of injuries by injury type in a sample of 1033 cases, showing increases in number of concussions by 400% and in skull fracture by 300% when the worker was not protected by a hard hat [15].

To reduce PPE violations in Korea, work safety measures and rules require workers not wearing a hard hat to be immediately dismissed from a construction site along with being fined and restricted from further participation to reduce PPE violations [16]. Likewise, to reduce casualties, governments and companies around the world have established

on-site safety policies and procedures to enhance construction site safety by educating the workers on the effectiveness of hard hats. However, according to the U.S. Occupational Safety and Health Administration (OSHA), non-use or inappropriate use of PPE is one of the most frequent violations of regulations in recent years [11,17,18]. Despite more than 90% of construction workers being aware of the importance of wearing hard hats, questionnaires have shown that discomfort or inconvenience are the main reasons for violations [19]. Currently, safety inspections regarding the use of hard hats are primarily reliant on inspector in-person (visual) monitoring. However, visual inspections are limited on construction sites, often due to the vast areas involved [20–22]. To supplement in-person inspections with limited human resources, a considerable number of studies have been conducted on automatic detection of hard hat use.

Vision-based methods are non-intrusive and can benefit from the broad application of surveillance cameras or closed-circuit televisions (CCTVs) [23]. CCTVs are often installed on modern construction sites for multiple purposes, because real-time video from CCTVs not only records construction progress but also provides a useful tool for investigating an accident; however, there are drawbacks of this system [24,25]. Most CCTVs or surveillance cameras are installed at a high altitude in fixed positions and their performance is heavily dependent on atmospheric conditions [26]. Moreover, supervisors have a high possibility of overlooking PPE violations or incidents where constant monitoring is required since construction sites often have more than one CCTV camera installed [27].

To overcome these limitations, researchers have applied object detection algorithms or deep learning methods for autonomous detection. Park et al. [24] presented a hard hat detection method based on a histogram of oriented gradients (HOG). With this method, human bodies and hard hats were initially detected by the cameras, and then, subsequent matching between the detected bodies and hard hats was performed based on their geometric and spatial relationships. In this manner, safety alerts could be issued automatically if the camera detected a human body without a hard hat. The field tests indicated greater than 94% overall precision, but these methods only detected standing or walking bodies and were unsuitable for bodies in various movement positions. Fang et al. [26] proposed an object recognition method based on Faster regions with convolutional neural networks (R-CNNs) for non-hard-hat use by using far-field surveillance videos on construction sites. The authors enabled real-time monitoring by adopting the Faster R-CNN method, which has the advantages of high precision and shorter calculation time. The results indicated that the proposed method could be effectively used to detect non-hard-hat use; however, the algorithm could not identify the workers involved. Moreover, the recall rate significantly dropped when only part of the head was visible. Wu and Zhao [27] presented a new approach for hard hat detection using a color-based hybrid descriptor. Similar to other methods, the authors aimed to extract features such as hard hats from videos using HOGs, which were developed to identify the colors of each hard hat by combining local binary patterns (LBP), Hu moment invariants (HMIs), and color histograms (CHs). Then, the authors applied a hierarchical support vector machine to classify the images based on hard hat color, which can impart different hard hat colors with different meanings on some sites. However, in many construction sites, hard hat colors are irrelevant, and workers often shift tasks.

In addition to previous studies, a number of relevant studies have been conducted recently to increase detection precision [28], to overcome construction environment complexities [29], and to analyze view variance [30].

The previously proposed methods of detecting hard hats have often required a significant amount of processing time to complete the analysis or have required close-up videos of a worker that had to be free of shadows, reflections, occlusion, etc., to achieve high precision [21]. As an alternative, as sensor technologies have improved, attempts have been made to control PPE usage based on commercially available radio frequency identification (RFID) tags. Researchers have demonstrated that RFID tags are suitable for indoor construction conditions due to their low cost, long operational life, and small

size but are unsuitable for outdoor environments due to the incremental signal attenu-
ation [20,31,32]. In addition, Park et al. [24] determined that even though using tags or
sensors on a construction site was cost effective, it was difficult to determine whether or
not the hard hats were being appropriately used.

Based on analysis of the existing approaches, this study assessed a novel method
to detect hard hat use with sensor technology. In this method, a commercially available
wearable smart band (PWB-300; Partron Co., Ltd., Hwaseong, Korea) [33] was modified by
changing the sensor placement from the arm to the hard hat chin strap and redesigning
the shape for user benefit. In this way, the sensor could automatically determine whether
a hard hat was being worn properly and was able to identify the current position of the
worker along with any abnormality in their heart rate.

## 2. Materials and Methods

### 2.1. Overview of the System Architecture

An overview of the proposed non-hard-hat use detection system is shown in Figure 1.
The transmitting part of the system, a modified version of the PWB-300 device, consisted
of two sensors (photoplethysmogram (PPG) and accelerometer (Acc)), two modules (low-
power wide-area (LoRa) and GPS), and a power supply. These were embedded into a
microcontroller unit (MCU). This method allowed for the installation of the PPG sensor
on the chinstrap of a hard hat. A hard hat can only offer protection when a worker wears
it properly. BS EN 397:1995 [34] specifies that all helmets/hard hats should be fitted with
a headband and chin straps with a width of not less than 10 mm. Even though most
construction workers are aware of the importance of wearing a hard hat, approximately
60% of construction workers have denied using chin straps due to discomfort, and this
often become the source of accidents [35].

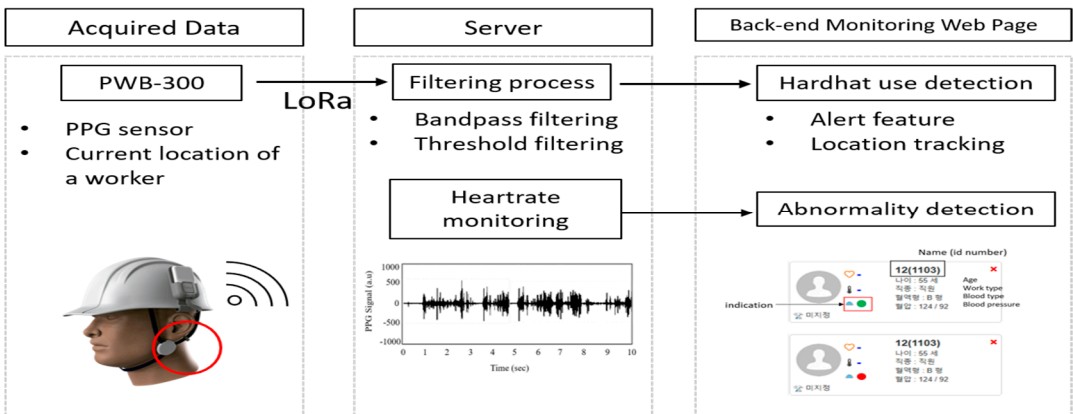

**Figure 1.** Overview of the non-hard-hat use detection system.

When a hard hat is properly worn and the device is initiated, the PPG sensor attached
to the chinstrap of the hard hat transmits data with one's location to the server via the
LoRa network. When the server receives the data, the monitoring webpage identifies the
hard hat user based on data from the PPG sensor. In this way, the manager is easily able to
verify the violation and the location of the worker by observing the webpage.

A monitoring function for an individual's heart rate was also added. If the server fails
to receive data from the PPG sensor or the received data show a heart rate abnormality, the
system sends a notification to the manager. The definition of "abnormality" was established
based on the results from previous studies [36–38]. The lower heart rate limit was 60 bpm,
while the upper limit was established based on age-based maximum heart rate formulas.
Among the various approaches, the following equation (Equation (1)) was selected due

to its high confidence interval bounds (±1 SD) and its suitability in a wide range of ages (30–75 years) [36].

$$\text{Maximum Heart Rate} = 207 - (0.7 * \text{age})\tag{1}$$

Accordingly, the flow diagram of the system is shown in Figure 2.

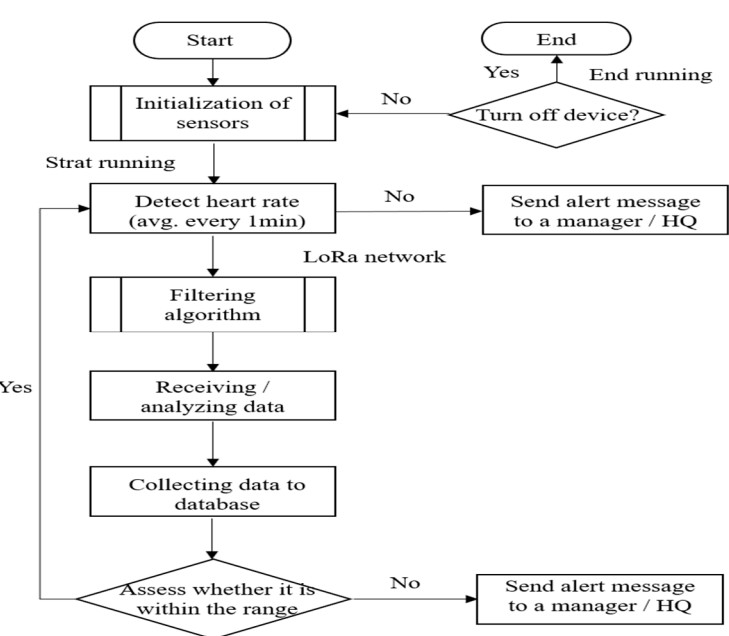

**Figure 2.** Flow chart of the overall system.

### 2.2. Hardware Part of the System

Figure 3 illustrates the components of the PWB-300, the transmitter node including two sensors (PPG and Acc), two modules (LoRa and GPS), and a power supply all embedded in an MCU. The receiver node, which consisted of the server that processed the data from the transmitter node, displayed the output on a mobile phone application or an Internet of Things (IoT) browser. Compared to other work environments, construction sites often are widespread and contain many environmental obstacles. Therefore, the LoRa network was chosen to connect the two nodes to provide high accuracy and wide coverage (up to tens of kilometers) at a low cost [39]. The technical specifications of the LoRa network module are shown in Table 1. Moreover, a help button was added for use in an emergency.

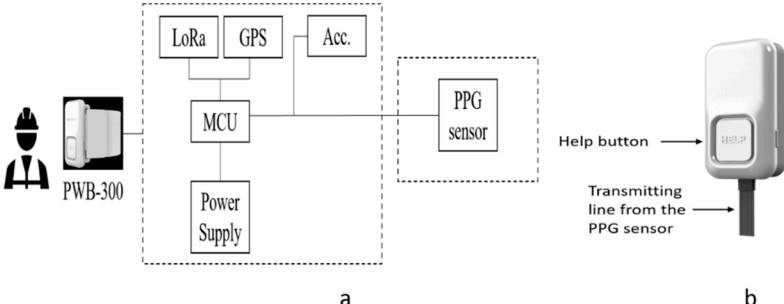

**Figure 3.** Hardware of the modified PWB-300: (**a**) component diagram of the device; (**b**) modified version of the device.

**Table 1.** Technical specifications of the low-power wide-area (LoRa) network.

| Specification | Value |
| --- | --- |
| Model | SX1276 (Semtech Corp, Camarillo, CA, USA) |
| Band | 920–925 MHz |
| Data rate | 0.18–37.5 kbps |
| Range | 5 km (urban); 15 km (rural) |
| Bandwidth | 7.8–500 kHz |
| Sensitivity | −111 to −148 dBm |

For the MCU, the model STM32L476 (STMicroelectronics international, N.V., Amsterdam, The Netherlands) was used, containing an ARM 32-bit Cortex-M4 CPU (Arm Limited, San Jose, CA, USA). An FPU with 512 KB flash memory and 129 KB SRAM was included. A PPG sensor (AFE4404; Texas Instruments, Dallas, TX, USA) with a dynamic range of 100 dB with 10 samples per second (SPS) at a 1000-SPS pulse frequency was used while an accelerometer (LSM6DS3; STMicroelectronics International, N.V., Amsterdam, The Netherlands) was applied. A GPS module (TD1030; Taidou Microelectronics, Shenzhen, China) with a 3 m horizontal and 4.5 m vertical accuracy was chosen. The power supply unit BQ24121 (Texas Instruments, Dallas, TX, USA) was selected for the power supply because it used a 3.7 V, 350 mAh, 502035 Li-ion polymer battery and had a 500-cycle battery life, which was greater than 80% of the initial battery capacity.

*2.3. Software Part of the System*

Back-End Monitoring Page

The ultimate goal of the developed monitoring page was to manage a worker's health and safety based on physiological data; however, only hard-hat-wearing detection was considered in this study. Each device contained its own MCU, which directly sent the data via the LoRa module in TCP/IP form, and an IoT gateway was not required in this system. The transmitted information was received by a back-end monitoring page where worker location and hard hat use were analyzed and displayed, as shown in Figure 4. A green light next to the hard hat symbol indicated that the worker was properly wearing the hard hat, and a red light indicated the opposite. Table 2 shows the specifications of the server.

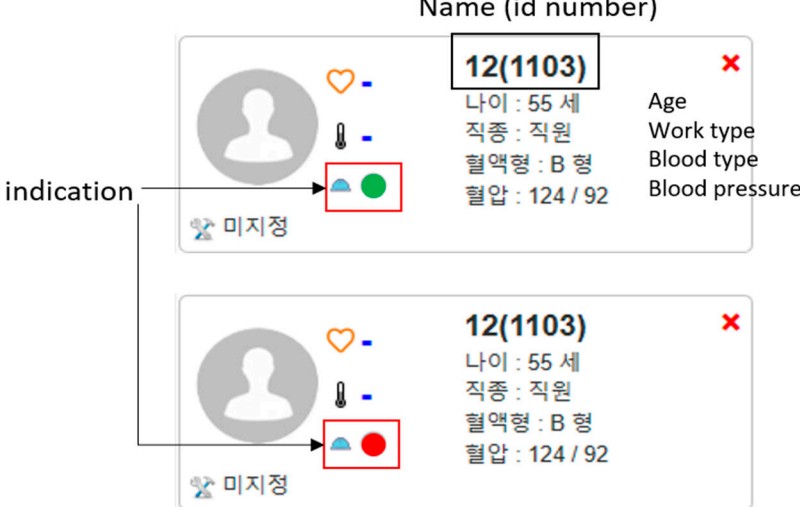

**Figure 4.** Hard hat use indication display.

**Table 2.** Technical specification of the server.

| Items | Details |
| --- | --- |
| Server | Lenovo SR530 (Lenovo, Beijing, China)<br>CPU: Xeon Silver 4208 8 core 85 W 2.1 GHz (Intel, Santa Clara, CA, USA)<br>RAM: 16 GB (1 × 16 GB) Single × 4 DDR4-2933 (Hewlett Packard Enterprise, San Jose, CA, USA)<br>HDD: 300 GB SAS 10 K 12 Gb *2 ea (Hewlett Packard Enterprise, San Jose, CA, USA) |
| Spectrum | 10 Mbps (1 Gbps Uplink) |
| OS | Server, OS, DB<br>Windows (IIS + MSSQL) |

As Figure 5 shows, the main page features simple tabs and visual data, which were designed to be accessible to the field manager. The IoT interface provides access to both PCs and smartphones, and the main features are as follows.

1.  Weather information: Shows the current worksite's atmospheric conditions by accessing the Korea Meteorological Administration (KMA) via the Internet.
2.  Status: Shows the list of registered workers on the site along with personal information including identification number, age, work type, blood type, normal heart rate, and hard hat use (Figure 4).
3.  Details: Shows the individual worker's transmitted data along with the worker's current location.
4.  Registration: A new worker can be registered by the field manager.
5.  History: Shows the individual's stored data from the server.

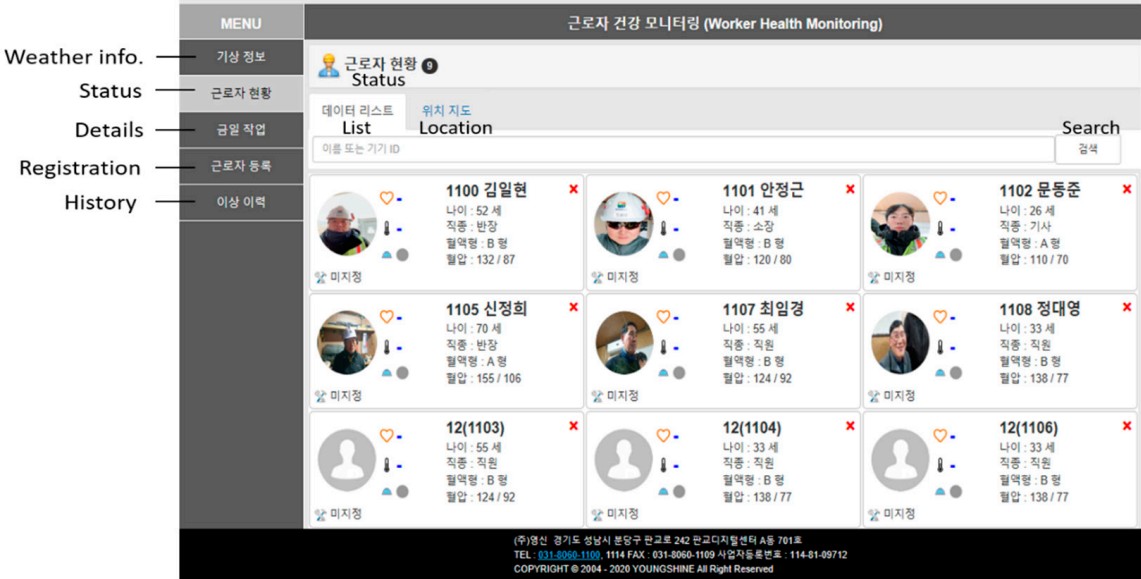

**Figure 5.** Back-end webpage interface of the system.

## 3. Sensor Design and Validation Test

To ensure that the proposed sensor was comfortable to wear, three different shapes of PPG sensors were tested. Each sensor was designed such that a user could position it for comfort, even though the three shapes and fixation mechanisms were different. In this study, the position and the shape of the sensor were studied through field testing. As Figure 6 illustrates, "Type A" and "Type B" were pin-shaped sensors that used a hook mechanism and sliding mechanism, respectively. "Type C" was a circular shaped sensor that used a sliding mechanism for positioning.

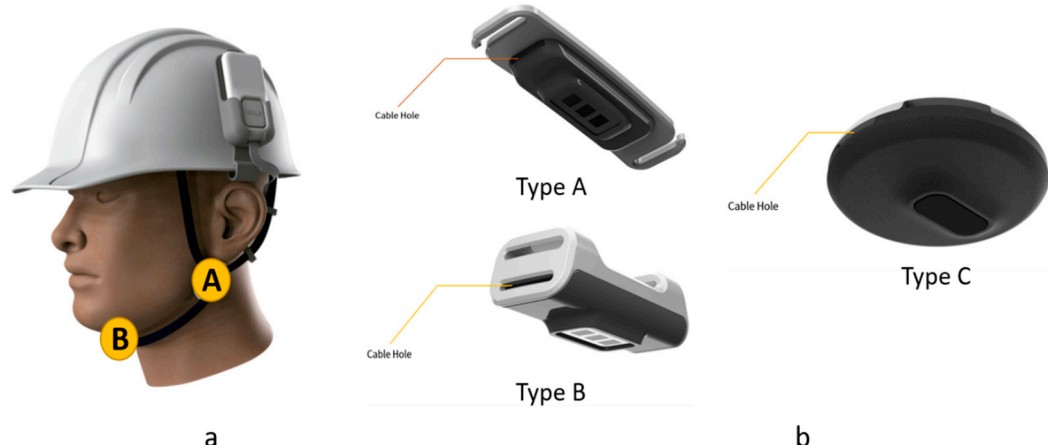

**Figure 6.** Location options and photoplethysmogram (PPG) sensor shapes: (**a**) considered sensor positions; (**b**) three shapes of PPG sensor.

For the performance comparison test, a large size construction site was chosen—the Gyeongbu Expressway Straightening Project, managed by Korea Expressway Corp. and Han-ra Corp., Korea. The test site was located in Wha-sung City, Korea and it was 4.70 km in length, with 10 lanes in each direction. The sizes were divided based on their costs of construction. Small construction sites involved construction costs less than USD 2 million, medium-sized sites used USD 2 million to 12 million, and those costing more than USD 12 million were large-sized construction sites. Each day, two randomly chosen frontline workers participated in this study for 3 h a day for a total of 15 days. One worker positioned the sensor at location A (near the ear), while the other worker positioned it at location B (under the chin). The two workers switched the location each day and they were asked to wear a different type of sensor—A, B, and C—each day. The technical specifications of the three types were identical, but the quality of the received data varied according to sensor shape. In addition, the percentage of data loss increased when the sensor was positioned at location B compared to when it was positioned at location A.

Table 3 shows the results of data loss from the three types of sensors and the two positions. The Type B sensor produced the most error, followed by Type A and Type C. Under normal thermal conditions, the shape of the sensor had a greater effect on data quality than did its position on the chin strap. The circular shaped sensors at location A showed the lowest error rate of 0.56%. Therefore, the Type C sensor was selected for this study (Figure 7). As Figure 8 shows, the pin-shaped sensors showed discontinuity of data readings due to worker movements that caused detachment from the skin or rotation of the sensor.

**Table 3.** Data loss by sensor type and position.

| Sensor Type/Location | 1 (Day) | 2 | 3 | 4 | 5 | 6 | 7 | 8 | 9 | 10 | 11 | 12 | 13 | 14 | 15 | Daily Avg in Min (Error Rate) |
|---|---|---|---|---|---|---|---|---|---|---|---|---|---|---|---|---|
| Type A/B | 18 (min) | 15 | 11 | 17 | 13 | 18 | 8 | 22 | 25 | 19 | 27 | 8 | 23 | 22 | 10 | 17.07 (9.48%) |
| Type A/A | 13 | 17 | 11 | 16 | 26 | 20 | 15 | 8 | 24 | 25 | 12 | 10 | 21 | 18 | 13 | 16.0 (8.89%) |
| Type B/B | 22 | 26 | 24 | 28 | 16 | 24 | 20 | 18 | 14 | 33 | 24 | 25 | 23 | 20 | 17 | 22.27 (12.37%) |
| Type B/A | 24 | 28 | 14 | 16 | 20 | 11 | 31 | 25 | 14 | 27 | 22 | 18 | 23 | 33 | 15 | 21.4 (11.89%) |
| Type C/B | 0 | 1 | 1 | 0 | 0 | 0 | 1 | 0 | 0 | 0 | 0 | 0 | 0 | 0 | 0 | 0.2 (1.67%) |
| Type C/A | 0 | 0 | 0 | 0 | 0 | 0 | 1 | 0 | 0 | 0 | 0 | 0 | 0 | 0 | 0 | 0.07 (0.56%) |

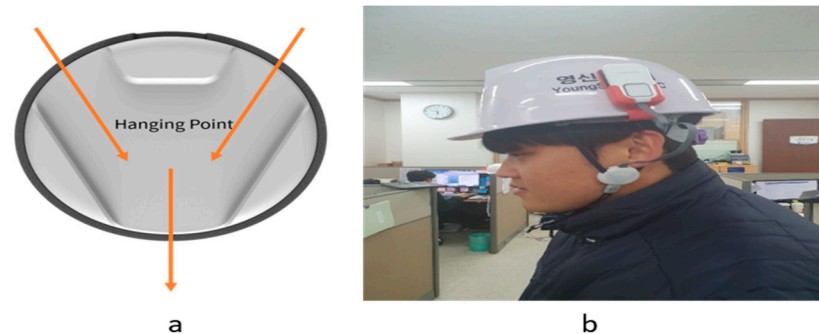

**Figure 7.** Type C PPG sensor: (**a**) the sliding design of the sensor; (**b**) prototype of the Type C sensor.

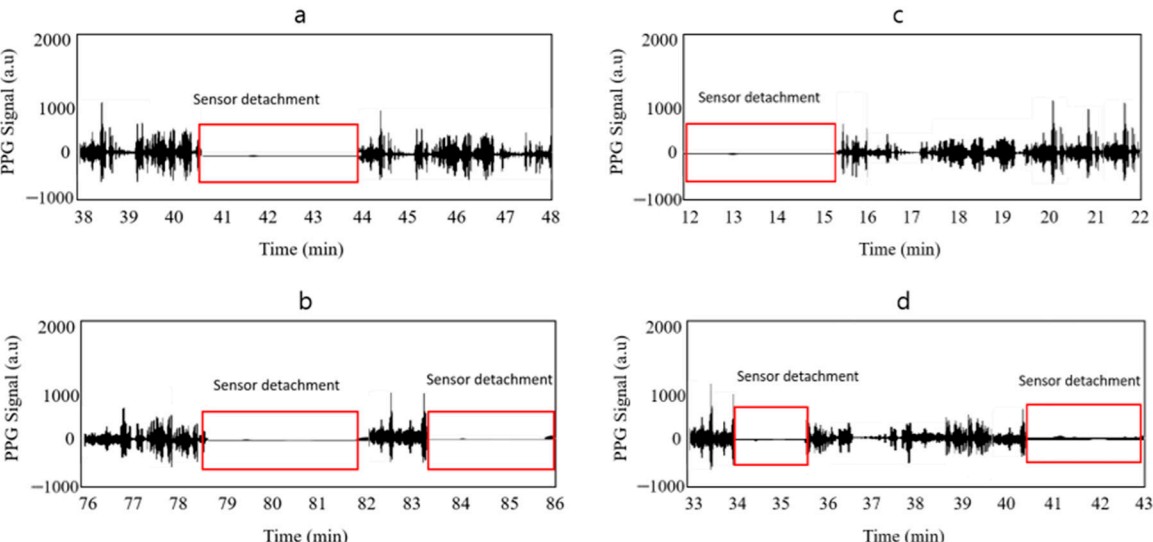

**Figure 8.** Samples of data loss: (**a**) PPG sensor Type A sensor detachment, (**b**) rotated sensor direction due to worker's movement, (**c**) PPG sensor Type B sensor detachment, and (**d**) rotated sensor direction due to worker's movement.

*Sensor Signal Filtering Process*

Because the selected device was located on the worker's head, the worker's movements could potentially cause an error in the readings. In addition, considering the typical construction site environment where a large number of noises and artifacts existed, noise removal and error correction processes were necessary. GPS data, which indicated the position of the worker, were proven to be accurate to about 3 to 5 m during the validation tests. Therefore, only the PPG sensor required correction. As shown in Figure 9, bandpass filtering and threshold filtering were used, while a range of 0.5–4 Hz was applied as a band filter to capture useful data [40].

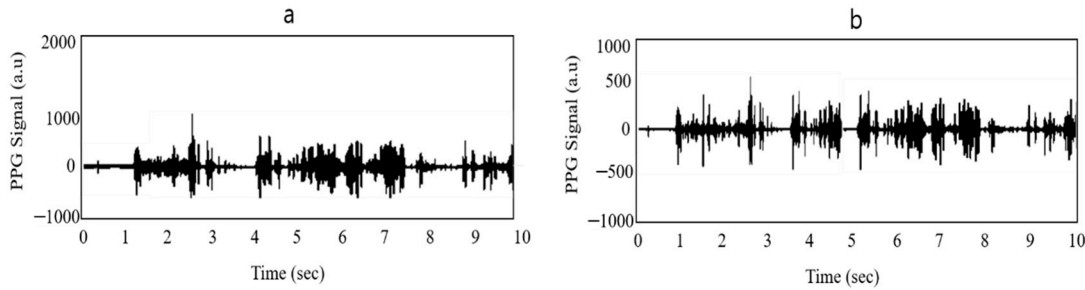

**Figure 9.** Signal filtering process: (**a**) before the filtering process; (**b**) after the filtering process.

## 4. Discussion

Despite management efforts to reduce the number of industrial accidents caused by PPE violations, construction workers give various excuses for not wearing hard hats properly, and they avoid detection by means of surveillance blind spots. Various approaches using sensor technology have been used to improve construction workers' health and safety using sensor technology [20,23,31,41,42], but previous attempts were aligned only with the intent or main purpose for which the sensor was developed. The novel method proposed in this study follows the conventional approach of monitoring a worker's heart rate through a PPG sensor and also detects hard hat non-use in a construction site without additional complicated equipment. To address the discomfort issues found in the previous studies [23,43] the developed method installs the sensor on the required chinstrap; three types of sensors were studied along with their optimal positions. The results indicated that pin-shaped sensors often cause chin straps to twist due to worker chin movements or actions. Therefore, data from that type were not constant and had an error rate of 9% to 12%. However, the validation tests were performed only with frontline workers, so workers with different tasks might show different results.

Sensor location on the chin strap was considered in this study, but location (near the ear and under the chin) did not have significant effects on the results under normal thermal conditions. The results showed 0.59%, 0.48%, and 1.11% differences between the two locations for Type A, B, and C sensors, respectively. According to validation tests, the GPS module in the device showed 3- to 5-m accuracy, which was considered acceptable since the results did not require pin-point location. Furthermore, in this test, the on-site manager was able to locate the worker who required attention, aided by transmitted GPS and registered worker information such as a facial photo. Moreover, the connection between the transmitting node and the receiving node in the LoRa network was effective on large construction sites. The quality of the data acquired from the validation tests enabled creation of a back-end monitoring webpage that is capable of detecting both heart rate abnormality and hard hat non-use.

## 5. Conclusions

This study developed an autonomous detection method based on sensor technology for hard hat non-use in a construction site. The modified design of a commercially available device in this study, the PWB-300 wearable smart band, overcame the major limitations of the existing vision-based detection methods in the construction industry.

The proposed method validated that hard hat use can be confirmed through a conventional sensor without extra equipment. Through field testing, optimal sensor shape and position on the chinstrap of a hard hat were identified. The circular shaped Type C sensor performed with the lowest error rate (0.56%), and the movements of the worker did not hinder the continuous transmission of data. In addition, location A (near the ear) showed an average of 0.73% better accuracy in data readings than location B (under the chin).

Moreover, using the transmitted data, a heart rate monitoring feature was added; if the server failed to receive data from the device or the transmitted PPG sensor data fell out of the age-adjusted heart rate range, the field manager received a notification. The current status of registered workers in the field can be monitored through the back-end monitoring page via the IoT or a smartphone, and a webpage was created to manage worker health and safety by monitoring individual physiological responses through the sensor technology. However, only the hard hat detection feature was considered in this study; the integration aspect will be addressed in a future study.

The current method was tested under normal thermal conditions, but future studies are needed to investigate the performance in different environmental conditions, along with the effects of worker sweat on the sensor.

**Author Contributions:** J.H.K. and B.W.J. conceptualized the study, and J.H.J., Y.S.L. and D.K.K. performed the investigations. J.H.K. developed the methodology and wrote the original draft of the manuscript. B.W.J. supervised the process while J.H.J., Y.S.L. and D.K.K. reviewed and edited the manuscript. All the authors have read the manuscript and approved the final version of the manuscript.

**Funding:** This research was financially supported by the Ministry of Trade, Industry and Energy (MOTIE) and the Korea Institute for Advancement of Technology (KIAT) through the Innovative Procurement-linked Technology Commercialization Program (P0013597—Development of Construction Site Safety Management Platform using the Smart Personal Protective Equipment).

**Institutional Review Board Statement:** Not applicable.

**Informed Consent Statement:** Informed consent was obtained from all subjects involved in the study.

**Data Availability Statement:** The data used to support the results of this study are available from the corresponding author upon request.

**Conflicts of Interest:** The authors declare that there is no conflict of interest regarding the publication of this paper.

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
