# Peer review of "Autonomous Detection System for Non-Hard-Hat Use at Construction Sites Using Sensor Technology"

_sustainability, doi:10.3390/su13031102_

Round 1

Reviewer 1 Report

The sensor cannot be “ sensor-based method” please adjust this

Not clear the link between “ sensor-based method” and “ transmitting node”

What does mean sensor “ PPG”..please define all the acronyms

The abstract is not clear difficult to take the idea that the authors wasn’t to send to readers  

As above I suggest to revise the entire work for English as there numerous phrases that are difficult to understand ..other example apart of the abstract are  listed below and other that are found everywhere

 “As the quality of life improves, the expectations for and interest in occupational safety are increasing, yet the accident rate in the construction industry still remains the highest “???

Who cause “ it still causes”??

You said “The overall sensor-based, non-hardhat use detection system is shown in Figure 1.” however in Figure 1 the caption indicate “Overview of the proposed detection system.” Which are two different things please make clear

“MCU.”???

“BS EN 397:1995 “ please cite this standard

What represent a “LoRa network”??

In Figure 4. You have to images what is the difefrnec between them ?

Figure 5. contain many Chinese/Korean words..is better convert them in English

How statistical representative is this study involving only “The two participants”

“various approaches have been tried” replace with “various approaches have been used”

The discussions are very brief please discus them in rapport to state of art

The conclusion should be more concise and it will be better to point out the achievements with some bullet points

Author Response

Comments and Suggestions for Authors

Comment 1: The sensor cannot be “sensor-based method” please adjust this

Answer: As reviewer commented, the title of the manuscript has been changed, and the term “sensor-based method” has been adjusted throughout the paper.

Comment 2: Not clear the link between “sensor-based method” and “transmitting node”

Answer: The transmitting node was used to imply data transmitting from the sensors. For the clarification, line 10-11 has been modified.

Comment 3: What does mean sensor “PPG”..please define all the acronyms

Answer: As the reviewer suggested, the acronyms for PPG has been defined (line 11) along with other technical terms throughout the paper.

Comment 4: The abstract is not clear difficult to take the idea that the authors wasn’t to send to readers

Answer: As reviewer commented, the overall abstract section was revised to deliver clear idea of the study. (line 9-23)

Comment 5: As above I suggest to revise the entire work for English as there numerous phrases that are difficult to understand ..other example apart of the abstract are  listed below and other that are found everywhere

Answer: As reviewer suggested, overall English of the paper has been revised.

Comment 6: “As the quality of life improves, the expectations for and interest in occupational safety are increasing, yet the accident rate in the construction industry still remains the highest “???

Who cause “ it still causes”??

Answer: Lines 27-29 has been modified for better understanding.

Comment 7: You said “The overall sensor-based, non-hardhat use detection system is shown in Figure 1.” however in Figure 1 the caption indicate “Overview of the proposed detection system.” Which are two different things please make clear

Answer: As reviewer commented, the line 588, and 600 has been modified for the clarification.

Comment 8: “MCU.”???

Answer: It is the acronym for microcontroller unit and the definition has been added in line 13.

Comment 9: “BS EN 397:1995 “ please cite this standard

Answer: Line 178-179 has been modified as reviewer suggested. (Citation 34)

Comment 10: What represent a “LoRa network”??

Answer: It represents low power wide area network. The definition for the acronym has been added in line 12.

Comment 11: In Figure 4. You have to images what is the difference between them ?

Answer: Figure 4 has been modified. The original intention was to show the differences in web display when a worker properly wear a hardhat and when a worker does not wear a hardhat properly.

Comment 12: Figure 5. contain many Chinese/Korean words..is better convert them in English

Answer: Figure 5 was modified according to the comment. Written Korean in Figure 5 was translated in English in Figure 4.

Comment 13: How statistical representative is this study involving only “The two participants”

Answer: Due to financial issue, this project allowed to produce two sets of three types of sensor. The authors agree that it would provide better statical results with more participants, however the authors concluded that the statistical results we received showed with more participants, the overall trend of the results will not vary significantly since two participants were chosen differently each day. For better understanding, line 854-862 has been modified.

Comment 14: “various approaches have been tried” replace with “various approaches have been used”

Answer: Line 1076-1077 has been modified as reviewer suggested.

Comment 15: The discussions are very brief please discus them in rapport to state of art

Answer: The overall discussion section has been modified in a manner that reviewer suggested. (Line 1074-1168)

Comment 16: The conclusion should be more concise and it will be better to point out the achievements with some bullet points

Answer: The overall conclusion section has been modified according to reviewer’s suggestion. (Line 1170-1191)

Reviewer 2 Report

I think the readers will appreciate the results of this manuscript.  Generally speaking, the manuscript is well written, the material is judiciously divided and organized and correct from scientific point of view. Some changes are, however, necessary. For these reasons I can recommend the acceptance of this paper after some corrections.

Before that the Editor makes a decision, I suggest that the authors emphasize take into account the following corrections

  1. In Introduction, author described past work, but little comment on the contribution and shortcoming. Author need to provide critical comments.
    2. Please highlight how the work advances or increments the field from the present state of knowledge and provide a clear justification for your work.
    3. In the conclusion, please show how the work advances the field from the present state of knowledge. Please provide a clear justification for your work in this section, and indicate uses and extensions if appropriate. Moreover, you can suggest future developments in the domain.
  2. The conclusion section has to be rewritten doing an effort to remark the main findings rather than summarizing the article content.

Author Response

Comments and Suggestions for Authors

I think the readers will appreciate the results of this manuscript.  Generally speaking, the manuscript is well written, the material is judiciously divided and organized and correct from scientific point of view. Some changes are, however, necessary. For these reasons I can recommend the acceptance of this paper after some corrections.

Comment 1: In Introduction, author described past work, but little comment on the contribution and shortcoming. Author need to provide critical comments.

Answer: As reviewer suggested, comments on the past work have been added in line 347-348, 353-355 , 361-363, 579-581.

Comment 2: Please highlight how the work advances or increments the field from the present state of knowledge and provide a clear justification for your work.

Answer: As reviewer suggested, discussion section has been modified to show the highlight of the proposed method (line 338-384)

Comment 3: In the conclusion, please show how the work advances the field from the present state of knowledge. Please provide a clear justification for your work in this section, and indicate uses and extensions if appropriate. Moreover, you can suggest future developments in the domain.

Answer: As reviewer suggested, overall discussion section has been modified and future studies have been added in conclusion section. (Line 1079-1204), (Line 1225-1227)

Comment 4: The conclusion section has to be rewritten doing an effort to remark the main findings rather than summarizing the article content.

Answer: The conclusion section has been modified to remark the main findings of this study. (Line 1206-1227)

Round 2

Reviewer 1 Report

The authors well responded to my comments and therefore I suggest acceptance.